# Attachment and trauma-informed programme to support forcibly displaced parents of youth in Sweden: feasibility and preliminary outcomes of the *eConnect Online* programme

Anna Kristen [1] , Raziye Salari [2] , Marlene Moretti,[1] Fatumo Osman [3]

¹Department of Psychology, Simon Fraser University, Burnaby, British Columbia, Canada
²Child Health and Parenting, Department of Public Health and Caring Sciences, Uppsala Universitet Institutionen for folkhalso- och vardvetenskap, Uppsala, Sweden
³School of Health and Welfare, Hogskolan Dalarna, Falun, Sweden

**Correspondence to**
Fatumo Osman; fos@du.se

## ABSTRACT

**Objectives** To assess the feasibility, acceptability and the impact of an online parenting programme for forcibly displaced parents of adolescents.

**Design** The study was a single-arm feasibility study using pre-intervention post-intervention and follow-up assessments.

**Setting** Participants were recruited from municipality-based activities for refugee parents in a small city in the south of Sweden.

**Participants** Participants were forcibly displaced parents (n=23; 47.8% maternal figures) of youth (n=23; 8–17 years old; 26.1% female) from Syria, Afghanistan and Somalia participating in an online parenting programme (*eConnect*).

**Intervention** *eConnect* is an attachment-based and trauma-informed parenting intervention and was delivered over the course of 10 weekly sessions.

**Primary and secondary outcome measures** Feasibility was assessed by programme enrolment, attendance, completion and acceptability of the online platform and cultural fit of the programme. Primary outcome measures were programme impact on youth mental health problems. Secondary outcome measures were programme impact on family functioning and parent–child attachment insecurity.

**Results** The *eConnect* programme was highly feasible in terms of overall enrolment (100%), attendance (89.6%) and retention rates (100%). The online platform was acceptable, with mixed feedback primarily related to the access and usage of technology. Cultural fit of the programme was acceptable. Youth mental health problems ($\eta^2$=0.29) and family functioning significantly improved ($\eta^2$=0.18) over the course of the programme. Unexpectedly, parent reports of youth attachment insecurity significantly worsened ($\eta^2$=0.16).

**Conclusions** The findings suggest that the online delivery of *Connect* was a promising way to reduce barriers to service access and improve mental health problems and family functioning among forcibly displaced parents and their children during COVID-19. Future research is needed to explore the acceptability and impact of this programme post-COVID-19, and to develop culturally tailored and psychometrically sound measures for parent and youth reports of attachment.

## STRENGTHS AND LIMITATIONS OF THIS STUDY

⇒ Use of mixed methodologies including quantitative and qualitative data.
⇒ High levels of participation and programme completion of culturally diverse parents in intervention study including 2 months follow-up.
⇒ Unique multicultural sample of forcibly displaced parents resettled in Sweden.
⇒ Small sample size and lack of control group.
⇒ Attachment parent-report measure requires validation in culturally diverse refugee samples.

## INTRODUCTION

Globally, we are witnessing unprecedented rates of forced displacement due to violence, conflict and human rights violations.[1] At the end of 2021, the forcibly displaced population exceeded 89 million, with over 40% being youth under the age of 18 years old.[1] In Sweden, there are currently over 250 000 forcibly displaced persons, many of whom have fled from ongoing conflicts in Syria, Afghanistan and Somalia.[2] Forcibly displaced families often experience significant pre-displacement trauma, including political conflict, war-related violence and dangerous circumstances during migration.[3] Upon migration, families face significant acculturative stress including unfamiliar social norms and expectations, experiences of systemic racism and discrimination and extreme deprivation of resources.[4 5] Familial acculturative stress is associated with poor family functioning,[6] disconnection within the parent–child relationship[7] and heightened youth vulnerability for poor mental health outcomes.[8]

Forcibly displaced youth are at high risk of developing mental health disorders.[9] Adolescence is a period of heightened

vulnerability for the onset of emotional and behaviour problems, and challenges related to assimilating or integration of cultural identity into their host culture is associated with poor mental health outcomes.[10] Indeed, a recent meta-analysis revealed that post-traumatic stress, depressive disorders and anxiety disorders are highly prevalent among refugee youth who have resettled in European countries (53%, 33% and 32%, respectively).[11]

Evidence-based parenting interventions are known to address youth mental health problems and enhance caregiving practices.[12 13] More specifically, emerging evidence has demonstrated that evidence-based parenting interventions adapted to address the unique needs of forcibly displaced parents may be effective in improving family outcomes such as reducing youth mental health problems[14 15] and promoting family cohesion.[16 17] Yet, forcibly displaced parents often experience exceptionally poor access to healthcare services, due to limited awareness of service availability, perceived stigma around mental health problems and fear of judgement, discrimination or child removal by service providers.[18] Service access is further complicated for these families by the limited availability of culturally aligned services delivered in their native languages.[19]

The COVID-19 pandemic exacerbated challenges with healthcare access for families across the globe. Low-income populations, including forcibly displaced families, were disproportionately impacted by the challenges of COVID-19 given the pre-existing barriers they experienced in accessing healthcare services.[20] In several countries, public health restrictions precipitated a rapid and widespread shift to virtually delivered healthcare services. Yet, we lack knowledge of how to develop and implement culturally and contextually tailored parent support services for underserved populations; the barriers and facilitators of programme uptake in these populations; and the impact of services in promoting parent, family and youth mental health.

Beyond COVID-19, virtually delivered interventions have been shown to overcome other key barriers to service access (eg, demanding work schedules).[21] Thus, the use of virtually delivered interventions may be a promising avenue to reduce service access disparity for underserved populations. A recent qualitative study exploring refugee experiences of virtually delivered mental health services highlighted that while access to technology posed a barrier to service use, online service delivery eliminated other barriers to service access (eg, transportation costs), and created new possibilities for culturally and contextually aligned service delivery in parents' native languages.[22] Despite the potential for virtually delivered services to improve service access for forcibly displaced communities, there is a complete absence of research examining the feasibility of virtually delivered parenting interventions for forcibly displaced families.

## The current study

The *Connect* programme is an attachment-based intervention designed for parents and alternative caregivers of youth aged 8–18 years old.[23] In previous studies, we adapted and implemented *Connect* for Somali-born parents in Sweden. The results demonstrated that the *Connect* programme was effective for improving both parental and youth mental health problems, and for improving parental efficacy and satisfaction, and these treatment gains were maintained at a 3-year follow-up.[15 17] The current study aims to assess the feasibility, acceptability and the impact of the online version of the *Connect* parenting programme (*eConnect Online*)[24] for forcibly displaced parents in Sweden.

## METHODS

### Design and study population

The current study is a part of a larger international research programme evaluating the impact of an online version of a parenting programme, *Connect*, on youth mental health during the COVID-19 pandemic.[24] The project included the preliminary evaluation of *eConnect* in a Canadian sample, and studies assessing the feasibility of the *eConnect* programme in culturally diverse contexts (eg, Mexico, South Africa). The current study was a single-arm feasibility study using pre-intervention, post-intervention and follow-up assessments. We recruited the parents from municipality-based activities serving refugee parents (eg, languages classes) in a small city in the south of Sweden from May until August 2021. Parents were invited to an informational session where they received information about the programme and the study. Both parents from families were encouraged to attend the programme. All parents with youth aged 8–18 years old were eligible to participate. We emphasised that participation in the research study was voluntary and would not impact participation in the *eConnect* programme. All parents completed written informed consent for participation.

### Patient and public involvement

Over the last 20 years, the *Connect* programme has been recursively shaped in response to families' feedback. In this study, continuous discussions were carried out with the municipality and Humana organisation for implementation of the *eConnect* programme. Additionally, parents were invited to describe their experience and provide feedback at the end of the programme.

### Intervention

*The Connect* programme is a manualised, attachment-based and trauma-informed intervention for caregivers of youth aged 8–18 years old.[23] Through the use of experiential roleplays and reflective exercises, the programme is designed to promote caregiver reflective functioning and sensitivity to strengthen the caregiver–teen relationship. Extensive research has provided evidence for the effectiveness of *Connect* in reducing serious youth mental

health problems.[12 25] The programme is delivered by two trained facilitators over the course of 10 weekly 90-min sessions.

*eConnect Online* was adopted given in-person gathering restrictions due to the COVID-19 pandemic. The online version of *Connect* mirrors all programme components (ie, principles, roleplays, reflective exercises, real-time flip charts) and real-time interactive group process components of the in-person version of the programme.[24] To support real-time interactions, *eConnect* uses a tech support person who supports parents' access to the online space and facilitates group exercises. The results from the preliminary evaluation of the *eConnect* programme in Canada were highly comparable to research evaluating the in-person method of delivery.[24]

Prior to the first session, the group facilitators held a technology orientation session in which parents were introduced to, and practised using, the videoconferencing platform. If needed, a limited numbers of computers were available for parents to borrow. In total, four *eConnect* facilitators (two women and two men) delivered the three groups included in this study. Three of the facilitators had previous experience in delivering the *Connect* programme in-person. All four facilitators had lived experience as forcibly displaced immigrants from Somalia, Iraq and Afghanistan.

In previous evaluations of the adaptation of the *Connect* programme for Somali-speaking parents in Sweden, the procedure included two initial group sessions focused on providing tailored societal information to parents such as information on the role of social services in Sweden.[15] In the current study, societal information was provided to newcomer parents prior to the *eConnect* programme. In this study, *eConnect* was delivered in Somali, Dari and Arabic languages. *eConnect* is flexible to cultural and contextual diversities. While the core attachment-based principles and components (eg, roleplays, reflection exercises) of *Connect* were retained, the group facilitators adjusted the roleplay content to align with the cultural context of each group.[26] For example, roleplay content was adjusted to capture conflict that commonly arises in displaced families related to acculturative tensions between child desire for cultural integration and autonomy and parental desire to uphold cultural values and practices, including respect for parental authority.[4] The facilitators also used metaphors and proverbs to explain aspects of attachment principles such as empathy and conflict. These modifications provided opportunities for parents to reflect on, discuss and practice different ways of integrating cultural beliefs and values with responsive and strength-focused parenting practices.

## Data collection procedure

Parents were invited to complete self-report measures at pre-intervention (T1), at the end of the programme (T2) and 2 months following the completion of the programme (T3). Most of the parents came to the municipality to complete the assessments where they were supported by group facilitators and interpreters. The lead researcher was also present via Microsoft Teams to support completion of research materials. Some of the parents completed research assessments at home and were encouraged to contact the lead researcher should any support be required. Parents were additionally invited to partake in focus group interviews (in their respective native languages) at the end of the programme, facilitated by the lead researcher via Microsoft Teams. The lead researcher has extensive experience in facilitating focus group interviews, and is fluent in Somali, Arabic, Swedish and English. A professional interpreter joined the Dari focus groups. To mitigate the potential for loss of meaning or accuracy in parents' descriptions during the interpretation process, the audio tapes were also reviewed for accuracy by a professional transcriber fluent in Dari. All parents received honorarium for completing research materials equivalent to US$10 at each timepoint.

## Feasibility and acceptability assessments

Feasibility of the intervention was assessed by programme enrolment, attendance and completion. For the current study, feasibility was determined by participant attendance of at least seven programme sessions. Acceptability of the virtual platform and cultural fit were assessed in the post-intervention group interviews. Parents additionally rated their familiarity before starting the programme with videoconferencing platforms at post-intervention.

## Outcome measurements

Primary outcomes were reductions in mental health problems in youth. The secondary outcomes included improved family functioning and reductions in parent–child attachment insecurity.

### Demographics

Parents also reported on sociodemographic information including their country of origin, age, highest level of education and employment status.

### Youth mental health problems

Youth mental health problems were assessed using the Brief Child and Family Phone Interview (BCFPI).[27] The BCFPI is a 39-item standardised caregiver-report screening tool used to assess youth emotional and behavioural problems. We used the externalising subscale (attention-deficit/hyperactivity disorder, oppositional defiant disorder and conduct disorder, six items each) and the internalising subscale (separation anxiety disorder, generalised anxiety disorder, major depressive disorder, six items each). We additionally calculated a total problem score (externalising and internalising scores combined). Parents rated the frequency of behaviours (pre-intervention baseline: over the past 6 months at T1; post-intervention: over the past 2 weeks at T2 and follow-up: over the past 2 months at T3). Items were rated on a 3-point Likert scale (1=never, 2=sometimes, 3=often). We additionally used the family functioning subscale that consists of six items rated on a 4-point Likert scale (1=strongly agree; 4=strongly

disagree). The BCFPI has excellent psychometric properties.[28] In this study, the subscales showed good internal consistency at pre-intervention (α≥0.81).

### Parent–child attachment insecurity

Parent–child attachment insecurity was assessed using the caregiver-report version of a brief form of the Adolescent Attachment Anxiety & Avoidance Inventory (AAAAI).[23] The AAAAI consists of two subscales assessing attachment anxiety (seven items) and attachment avoidance (nine items). Items are scored on a 7-point Likert scale (1=strongly disagree to 7=strongly agree). We additionally calculated a total attachment insecurity score (anxiety and avoidance scores combined). Factor analytic research on the AAAAI has demonstrated strong psychometric properties.[23] In the current study, internal consistencies were acceptable (α ranging from 0.66 to 0.75).

### Statistical analyses

Statistical analyses were conducted using SPSS V.27.[29] Descriptive statistics were used for summarising programme enrolment, attendance and completion rates (n=30), and pre-intervention demographic data (n=23). To assess intervention effect on outcomes, one-way repeated measure analysis of variance (ANOVA) was conducted to determine within-subject change over time from pre-intervention to post-intervention and 2-month follow-up. Of the 23 parents that consented to complete the study measures, one did not complete the follow-up assessment and was excluded from these analyses. We first conducted omnibus tests to determine whether there were significant mean differences overall between the timepoints for the study outcomes. The effect sizes of the overall differences were determined using an eta-squared calculation ($\eta^2$=0.01–0.06 small effect, $\eta^2$=0.06-0.14 medium effect, $\eta^2$>0.14 large effect). We calculated the CIs for effect sizes using the method suggested by Wuensch.[30] When omnibus tests were significant (p<0.05), we tested for patterns of change (ie, linear and quadratic) over time.

## RESULTS

### Participant demographics

Of the 30 parents who attended the informational session, all parents enrolled in the *eConnect Online* programme, including two couples. Three programme groups were completed between September and December 2021: one for Somali-speaking parents, one for Dari-speaking parents and one for Arabic-speaking parents. We collected session attendance and programme completion rates for all 30 parents. However, to avoid dependency in the data, only one parent from each couple was eligible to complete the study outcome measures. Of the 28 parents eligible to participate in the research, 23 (82.1%) consented and were retained to complete study outcome measures.

The retained sample was biological parents (n=23; 47.8% maternal figures; aged 28–64) of adolescents

**Table 1** Pre-intervention demographic characteristics of sample

| Variables | |
|---|---|
| Parents, n (%) | |
| Mothers | 11 (47.8) |
| Fathers | 12 (52.2) |
| Participant age in years (mean±SD) | 47.09±8.34 |
| Years spent in host country (mean±SD) | 7.63±7.58 |
| Highest education level, n (%) | |
| No formal education | 5 (22.7) |
| Partial/completed primary school | 9 (40.9) |
| Partial/completed high school | 6 (27.2) |
| Apprenticeship/trades certificate | 1 (4.5) |
| College/university completion | 1 (4.5) |
| Financial well-being, n (%) | |
| Not enough money to cover living expenses | 2 (9.1) |
| Barely enough money to cover living expenses | 12 (54.4) |
| Enough money to cover living expenses | 8 (36.4) |
| Civic status, n (%) | |
| Married | 21 (91.3) |
| Single | 2 (8.7) |
| Employment status: unemployed, n (%) | 18 (90) |
| Multiple caregiver homes, n (%) | 19 (82.6) |
| No. of children in house (mean±SD) | 4.70±2.03 |
| Child sex: female, n (%) | 6 (26.1) |
| Child age (mean±SD) | 13.22±2.56 |
| Child living in the parents' home, n (%) | 23 (100) |
| Child attending regular school, n (%) | 23 (100) |

(n=23; 26.1% female; aged 8–17) who were forcibly displaced and living in Sweden. The parents reported their country of origins to be Afghanistan (n=10; 43.5%), Somalia (n=8; 34.8%) and Syria (n=5; 21.7%). Table 1 outlines additional demographic information.

### Feasibility: programme uptake and completion

Across all three groups—for Somali-speaking, Dari-speaking and Arabic-speaking parents—all parents who were invited to enrol in the programme (n=30) attended between 7 and 10 sessions (M=8.87, SD=1.06) and completed the programme (100%), showing excellent programme uptake across all three groups.

### Acceptability of the online platform
#### Online delivery

We intended to deliver the groups online, with parents and facilitators joining from their respective locations on their respective computers. In the Arabic group, the programme ran online as anticipated. The Somali group also ran online as anticipated; most joined from home except two parents who joined on their computers from

the municipality due to internet instability at home and challenges using the videoconferencing platform. For the Dari group, adjustments to the delivery method had to be made after the first session. The online delivery was adjusted to a hybrid delivery due to parents' challenges with accessing stable internet and necessary technology. In this group, all parents attended in-person at the municipality office, and the two group facilitators joined online.

### Familiarity with technology and accessibility of online delivery

At post-intervention, 17 parents (74%) rated their level of familiarity with videoconferencing platforms prior to beginning the group. Of these, 50% were somewhat familiar with videoconferencing, 11.1% had used videoconferencing platforms before but were not familiar with the technology and 38.9% had never used videoconferencing platforms before.

In the post-intervention focus group interview, all parents (n=23) reported about their experiences with the online programme delivery, including the advantages and disadvantages. Overall, the Arabic-speaking group (n=5) shared positive feedback, reporting on the many advantages of the online programme delivery. For this group, the online delivery facilitated the attendance of both parents from each family and eliminated stressors related to being away from the home (eg, childcare). The parents also enjoyed the comfort that came with participating from their home (eg, having a cup of tea). The group reported that the online delivery of the roleplays was excellent, and the online delivery did not interfere with feeling 'seen' by the group facilitators.

From the Somali-speaking group (n=8), some parents reported technology problems that sometimes made it challenging to attend the group. One parent reported feeling confused by the technology components. In addition, some parents reported that they would have preferred the interactions and connections that comes with being in-person. Despite technology challenges, the parents reported that they were still able to fully engage in the group discussions. One participant acknowledged that without the option for online delivery, attending the programme would not have been possible given the circumstances of COVID-19.

The Dari-speaking group (n=10) reported a number of technology challenges. Some of the parents were unable to access a computer or stable internet. Those that were able to access the necessary technology reported difficulties using the videoconferencing platform. The parents reported having no previous experience in using online learning methods. The parents reported that being together in-person was central to their learning. For example, the parents noted that distractions at home (eg, children) could get in the way of engaging in the programme and retaining information. Additionally, they noted the advantages of being in-person in terms of connecting with one another. The parents agreed that the roleplays delivered online by facilitators were important

to their learning, and the delivery of this was not impeded by the online platform.

### Acceptability of programme cultural fit

Parents additionally provided feedback regarding the cultural fit of the programme during the post-intervention focus group interviews. All three groups reported that it was essential that the group facilitators had lived experiences as immigrants and as parents. The parents explained that having group facilitators support them and truly understand their concerns as parents was contingent on the group facilitators having experienced the pain and challenges themselves. This lived experience was also essential for the group facilitators to accurately adapt the roleplays to the everyday challenges that the parents encounter at home with their teenagers, and all three groups reported that the roleplays did capture these challenges.

### Intervention outcomes

The within-subject changes from pre-intervention to post-intervention and 2-month follow-up are presented in table 2.

### Youth mental health problems

For parent reports of total problems on the BCFPI, the repeated measures ANOVA revealed an overall significant effect with large effect sizes ($\eta^2=0.29$; 95% CI (0.09 to 0.43)). There were both significant linear and quadric components, indicating that total problems significantly decreased from pre-intervention (M=9.36, SD=2.11) to post-intervention (M=7.85, SD=1.61), and then remained almost the same from post-intervention to 2-month follow-up (M=7.88, SD=1.76).

A similar pattern of results was found for youth externalising problems, with large effect sizes ($\eta^2=0.27$; 95% CI (0.07 to 0.41)). Parents reported fewer externalising problems overtime (M=4.83, SD=1.36; M=3.73, SD=0.92; M=3.92, SD=0.10, over the three timepoints, respectively), but the greatest reduction occurred from pre-intervention to post-intervention. For internalising behaviours, the repeated measure ANOVA revealed no significant overall changes across the three timepoints.

### Family functioning

The repeated measures ANOVA revealed an overall significant effect on family functioning, with large effect sizes ($\eta^2=0.18$; 95% CI (0.02 to 0.32)). The linear component was significant while the quadric component was not, indicating that family functioning improved linearly over time from pre-intervention (M=1.64, SD=0.64) to post-intervention (M=1.34, SD=0.25) and from post-intervention to 2-month follow-up (M=1.27, SD=0.34).

### Parent–child attachment insecurity

The repeated measures ANOVA revealed an overall significant effect on parent–child attachment ($\eta^2=0.16$; 95% CI (0.01 to 0.30)). The linear component was significant while the quadric component was not. Counter to

**Table 2** Within-group comparison of programme parents at pre-intervention, post-intervention and 2-month follow-up (n=22)

| Outcome measure | Pre-intervention, mean (SD) | Post-intervention, mean (SD) | Follow-up, mean (SD) | Overall | | Linear | | Quadratic | |
|---|---|---|---|---|---|---|---|---|---|
| | | | | F(2,42) | $\eta^2$ | F(1,21) | $\eta^2$ | F(1,21) | $\eta^2$ |
| BCFPI | | | | | | | | | |
| Externalising | 4.83 (1.36) | 3.73 (0.92) | 3.92 (0.10) | 7.63* | 0.27 | 9.88** | 0.32 | 5.71* | 0.21 |
| Internalising | 4.52 (1.06) | 4.12 (1.13) | 3.95 (0.94) | 2.53 | 0.11 | – | – | – | – |
| Total difficulties | 9.36 (2.11) | 7.85 (1.61) | 7.88 (1.76) | 8.60** | 0.29 | 10.24** | 0.33 | 5.94* | 0.22 |
| Family functioning | 1.64 (0.64) | 1.34 (0.25) | 1.27 (0.34) | 4.53* | 0.18 | 5.46* | 0.21 | 1.90 | 0.08 |
| AAAAI | | | | | | | | | |
| Avoidance | 2.02 (0.84) | 2.40 (1.02) | 2.39 (1.02) | 1.21 | 0.06 | – | – | – | – |
| Anxiety | 3.07 (1.27) | 3.58 (1.47) | 3.97 (1.57) | 3.91* | 0.16 | 6.98* | 0.25 | 0.04 | 0.00 |
| Insecurity | 2.55 (0.79) | 2.99 (1.00) | 3.18 (1.06) | 4.12* | 0.16 | 6.09* | 0.23 | 0.55 | 0.03 |

*p<0.05, **p<0.01. Lower scores on the BCFPI subscales indicate reductions in internalising and externalising problems, and improved family functioning. Lower scores on the AAAAI subscales indicate reductions in attachment insecurity (anxiety and avoidance).
AAAAI, Adolescent Attachment Anxiety & Avoidance Inventory; BCFPI, Brief Child and Family Phone Interview.

predictions and prior research, results showed attachment insecurity worsened linearly over time from pre-intervention (M=2.55, SD=0.79) to post-intervention (M=2.99, SD=1.00) to 2-month follow-up (M=3.18, SD=1.06). A similar pattern of results was found for parent–youth attachment anxiety subscale ($\eta^2$=0.16; 95% CI (0.01 to 0.30)), with large effect sizes. Parents report steady increase in attachment anxiety over the course of the intervention from pre-intervention (M=3.07, SD=1.27) to post-intervention (M=3.58, SD=1.47), and to 2-month follow-up (M=3.97, SD=1.57). There were no significant overall changes in attachment avoidance across the three timepoints.

## DISCUSSION

This single-arm study was conducted to test the feasibility, acceptability and impact of an online attachment-based programme for forcibly displaced parents of adolescents. We found that the *eConnect* programme was feasible with strong uptake reflected in enrolment (100%), excellent attendance (89.6% of sessions completed) and exceptional programme completion for all parents (100%) across the three groups. Parent feedback indicated *eConnect* enabled both parents from families to attend the sessions. More than 50% of the parents were fathers. This is a notable finding as fathers are typically under-represented across parenting interventions,[31 32] despite the critical impact that father parenting practices have on positive youth mental health.[33] Prior research has also called for the inclusion of both parents in intervention work whenever possible to target multiple risk factors related to youth adverse outcomes (ie, mother and father parenting practices).[34] The *eConnect* programme also reduced barriers to service access and eliminated obstacles related to being away from home (eg, childcare), which is of importance given the significant barriers to service access that forcibly displaced families face.[18]

The online delivery of the programme was acceptable to most of the parents. Although *eConnect* offered a means for forcibly displaced people to access parenting support during COVID-19, a lack of access to technology (ie, computers and stable internet) as well as lack of familiarity with using videoconferencing platforms was a major challenge for some parents, primarily the Dari-speaking parents. In our study, the Dari-speaking parents were generally newer migrants, compared with the other groups, and thus they likely had less exposure to technology through the Swedish labour market. Indeed, the majority Dari-speaking parents reported that they had never used a videoconferencing platform before.

In terms of lack of access to technology, previous research conducted during COVID-19 has shown that lower household income is associated with lower levels of digital literacy and lower rates of access to virtual healthcare.[22 35] Our findings, and the high rates of poverty among forcibly displaced families, underscore the need for service provider support in facilitating access to technology for parents, and support in increasing parental familiarity with videoconferencing platforms for successful implementation. This seems particularly crucial for displaced parents that are newer to their host country.

Despite these technology challenges, the high rates of programme enrolment, attendance and completion emphasise that parents were able to engage in the programme, indicating that *eConnect* is a promising avenue for reducing service access disparity. Our findings additionally emphasise the importance of shared language and lived experience between the group facilitators and participants for parental engagement. This finding might call attention to the need for culturally sensitive trainings for facilitators from non-migrant backgrounds for effective delivery. The current findings reveal the pressing need and eagerness among

forcibly displaced parents to address complex caregiving challenges. Importantly, the online delivery the *Connect* programme makes it possible for municipalities or organisations to use facilitators who are geographically distant, increasing programme accessibility and cost-efficiency.

We also completed a preliminary evaluation of the impact of *eConnect* Online on youth mental health problems and family functioning. Consistent with recent research evaluating *eConnect* in Canada and other culturally diverse contexts,[23 36] our findings indicated that the youths' total mental health problems, including externalising problems, significantly declined from pre-intervention to post-intervention and therapeutic gains were maintained at the 2-month follow-up. While youths' internalising problems did not significantly decline, the mean change scores suggest a reduction in internalising problems. This finding could reflect our small sample size or the less observable nature of internalising symptoms,[37] compared with externalising symptoms. This finding was also present in our previous study,[15] in which we did not detect a significant change in internalising problems from pre-intervention and 2-month follow-up but did observe a significant change in the long-term follow-up (3 years).[17] Consistent with prior research, family functioning significantly improved from pre-intervention to post-intervention, and post-intervention to 2-month follow-up.

Unexpectedly, our findings showed parents reported significantly higher rather than lower levels of attachment insecurity over the course of the intervention. This is surprising in light of the significant improvements in youth mental health symptoms and family functioning (eg, emotional closeness, openness in communication), and inconsistent with previous studies showing reductions in parent and youth reported levels of attachment insecurity following participation in the *Connect*[12 38] and *eConnect* programme.[23 36] It is important to note that the parent-report measure of attachment used in this study has not been previously validated within the culturally diverse and forcibly displaced families who participated in this research. Even though the measure was translated into the different languages of parents in this study, anecdotally it was observed that most parents struggled to understand the meaning of the items. It is possible that the meaning of these items differs for parents from these cultures due to differences in parenting goals and family values (eg, emphasis on youth independence vs family interdependence).[39] Alternatively, it is possible that forcibly displaced parents focus foremost on their child's tangible needs, risks and safety given the magnitude of their displacement experiences. Youth themselves may also mask feelings of distress as not to further burden their parents. Qualitative research may be needed to support a better understanding of the language and concepts that capture parent–child attachment for forcibly displaced parents, and the complex processes that may unfold in parent–child communication due to trauma.

## Strengths and limitations

Our feasibility study has several limitations. First, while all the parents completed the programme, our sample was small, and we had no control group. Second, even though all families who participated in the study shared a common history of forcible displacement, there may have been differences between their cultural backgrounds that we were unable to explore due to our small sample. Third, even though the majority of the eligible parents (82%) consented to completing study outcome measures, we were not able to collect demographic information from the parents who did not consent. However, all parents recruited to participate in the programme met the same inclusion criteria including parenting status, age of child, history of forcible displacement, and all parents were either Somali, Dari or Arabic speaking, so it is unlikely that those who participated or declined to do so differed in significant ways. Thus, despite our small sample size, we consider this research to be a valuable first step, given that forcibly displaced populations are greatly under-represented in intervention research. Finally, the measure of youth mental health problems (BCFPI) and the attachment measure (AAAAI) used in this study were not validated for this population. However, the internal consistencies in this study showed acceptable levels. There is a significant need for future research to deeply examine how forcibly displaced families conceptualise and understand attachment to inform the development and adaptation of self-report attachment measures, and to investigate processes that underlie the impact of displacement trauma on parents, youth and attachment. Future research should aim to include the voices of both parent and youth reports in both qualitative and quantitative methods of data collection. Despite these limitations, this is the first study to our knowledge to evaluate the online delivery of a parenting intervention for forcibly displaced parents. The positive changes in youth mental health can also be seen as strength, given the high rates of mental health problems and high parental stress prior to and during COVID-19.[40]

## Conclusion

The findings suggest that *eConnect* is a promising way to reduce barriers to service access and improve youth mental health problems and family functioning beyond COVID-19. This study underscores the important logistical considerations for the implementation of *eConnect* for forcibly displaced families, particularly the importance for service providers to consider how to support parents in accessing and feeling confident using the necessary technology. Service providers should also consider the expressed importance of the cultural and lived experiences of facilitators. Our findings emphasise the need for research to explore how forcibly displaced families conceptualise parent–child attachment in order to inform the development or adaptation of self-report measures for this population.

**Acknowledgements** This work was completed with the support of a Canada Research Chair - Tier 1 in Youth Clinical Psychological Science. The authors would like to thank all the parents who participated in this study. We also would like to thank all the group facilitators who delivered the intervention and assisted us with the recruitment. Finally, we thank the Humana and Växjö municipality for the technical support.

**Contributors** FO and MM conceptualised the study. MM is the developer of the Connect program. FO designed the study and led data collection. AK prepared the study data, and AK and RS performed the statistical analysis. All authors contributed to the interpretation of results. AK wrote the first draft of the manuscript. FO, RS and MM contributed to critical revision of the manuscript. All authors approved the final version before submission. The corresponding author attests that all listed authors meet authorship criteria and that no other authors meeting the criteria have been omitted. FO is the guarantor of the study.

**Funding** This work was supported by the Canadian Institutes of Health Research grant number #448851.

**Competing interests** None declared.

**Patient and public involvement** Patients and/or the public were involved in the design, or conduct, or reporting, or dissemination plans of this research. Refer to the Methods section for further details.

**Patient consent for publication** Not applicable.

**Ethics approval** This study involves human participants and was approved by Ethical approval was received from the Swedish Ethical Review Authority (Dnr. 2020-06737) and Simon Fraser University Office of Research Ethics (#2011s0284 and #20200401). Participants gave informed consent to participate in the study before taking part.

**Provenance and peer review** Not commissioned; externally peer reviewed.

**Data availability statement** Data are available upon reasonable request. The data that support the findings of this study are available on request from the corresponding author. The data are not publicly available due to privacy or ethical restrictions.

**ORCID iDs**
Anna Kristen http://orcid.org/0000-0002-6020-7711
Raziye Salari http://orcid.org/0000-0002-6363-4030
Fatumo Osman http://orcid.org/0000-0002-2038-9402

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
