## [Reviewer comments · BMJ Open]

ARTICLE DETAILS

TITLE (PROVISIONAL)	An Attachment and Trauma Informed Program to Support Forcibly Displaced Parents of Youth in Sweden: Feasibility and Preliminary Outcomes of the eConnect Online Program.
AUTHORS	Kristen, Anna; Salari, Raziye; Moretti, Marlene; Osman, Fatumo

VERSION 1 – REVIEW

REVIEWER	BARONE, LAVINIA Università degli Studi di Pavia, Brain and Behavioral Sciences
REVIEW RETURNED	11-Mar-2023

GENERAL COMMENTS	I read with interest the paper titled “Delivering an Online Attachment and Trauma Focused Parenting Support Program for Forcibly Displaced Parents: Adaptations and Feasibility”, given the seminal contribution of this topic in parenting support programs scientific literature: The Authors addressed an important topic and tried to contribute with the goal of testing the possible adaptation and feasibility of an online parenting program eCONNECT. The contribution is of value even for the special population treated, i.e., forcibly displaced parents with a trauma history. I have a few remarks which constitute minor revisions and that are listed below: p. 2 abstract sample description of both parents and youths could also include gender. Introduction Is clear well addressed and consistent with study aims. Aims of the study need to be reported at the end of introduction. Methods Design and study population p. 6 please provide a full description of the sample enrolled (which is reported only in the results section), including how many were couples and thus non independent data for analyses. Also, important to know the percentage of attendance for the whole group of parents. Did the Authors treated the data as non-independent when required, because the two parents of a couple rated the same child? This point needs to be addressed
---

	Results P 13 did the parents who didn't accept research enrollment differed from the parents who accepted (N = 23) on pre-intervention demographic characteristics? This info is important as feasibility research is being carried out. Discussion I found the discussion well written, clear and effective in addressing research aims, thus contributing to the literature research in a very challenging family context. I would add similar and recent research implementing eCONNECT in different context, e.g. Benzi I. et al., 2023 International Journal of Environmental Research and Public Health, 20, 3532. 1-11. https://doi.org/10.3390/ijerph20043532, for a more complete discussion References Some reference need to be checked e.g the year of Lin, B., & Moretti, M. M. (2022; in press) eConnect: Implementation and preliminary evaluation of a virtually delivered attachment-based parenting intervention during COVID-19. Attachment & Human Development To sum up my comments, I found the topic addressed of great value and the paper interesting and with minor revisions for publication
--	--

REVIEWER	Janowski, Roselinde Oxford University
REVIEW RETURNED	08-May-2023

GENERAL COMMENTS	Thank you for providing me with the opportunity to review your feasibility study of eConnect for forcibly displaced parents of adolescents in Sweden. Findings from this study will offer important lessons for implementing culturally appropriate and acceptable digital parenting interventions for forcibly discaled families. The manuscript is well written. I only have a few minor comments. 1. Minor proof-reading needed: for instance, on page 4, line 33, the word "period" had been written twice in the same sentence. 2. Data collection: 2.1. On page 9, line 6, the authors state that most parents came to the municipality to complete assessments. Where was the data collected for the parents who did not come to the municipality and who assisted this data collection? 2.2. On page 9, line 15, the authors state that the focus group interviews were facilitated by the lead researcher. Please provide a bit more detail surrounding this decision. Does the lead researcher speak the local languages of the participants? If no, were there any translators to assist the process to ensure that participants understood the questions? If the focus groups were not conducted in the local languages of participants, it may also be worth
--

	acknowledging this as a limitation in the discussion (being reflexive of the culture, race, and language of the interviewers). 3. Statistical analysis: 3.1. On page 12, line 20, the authors describe the effect size used in the analysis. It would be beneficial for interpretation of the results if 95% confidence intervals were also included. 3.2. It would be helpful to understand how many of the 23 parents were participating with their partner, i.e., in a couple. The authors should consider how this might affect 1) the interpretation of engagement results and 2) their analysis of change. That is, it may not be enough to merely use ANOVAs to determine within-subject change when individuals are nested within a couple/family. 4. Outcome measures: Have the primary and secondary outcomes been tested and/or in refugee populations?
--	---

VERSION 1 – AUTHOR RESPONSE

Reviewer 1, Comment 1: I read with interest the paper titled “Delivering an Online Attachment and Trauma Focused Parenting Support Program for Forcibly Displaced Parents: Adaptations and Feasibility.” Given the seminal contribution of this topic in parenting support programs scientific literature. The Authors addressed an important topic and tried to contribute with the goal of testing the possible adaptation and feasibility of an online parenting program eCONNECT. The contribution is of value even for the special population treated, i.e., forcibly displaced parents with a trauma history.

Author reply: We thank the author for noting the value of intervention research with this population.

Reviewer 1, Comment 2: P. 2 abstract: sample description of both parents and youths could also include gender.

Author reply: As suggested, we have included both parent and youth gender in the abstract (pg. 2).

Reviewer 1, Comment 3: Introduction is clear well addressed and consistent with study aims. Aims of the study need to be reported at the end of introduction.

Author reply: The aims of the study have been reported at the end of the introduction under the current study header (pg. 6).

Reviewer 1, Comment 4A: Design and study population: p. 6 please provide a full description of the sample enrolled (which is reported only in the results section), including how many were couples and thus non independent data for analyses.

Author reply: As suggested, we have included the number of couples participating in the program and we have clarified that only one parent from each couple was eligible to participate in the study to avoid dependency in the data (pg. 12). We appreciate the reviewers’ suggestion to provide the sample description in the methods. However, to align with the reporting guidelines set out by BMJ Open, we have structured the manuscript based on the Strengthening the Reporting of Observational Studies in Epidemiology (STROBE) checklist for cohort studies (link: <https://www.strobe-statement.org/checklists/>). Thus, we are inclined to retain the sample descriptions in the methods. We are happy to include these details in the manuscript if requested by the action editor.

Reviewer 1, Comment 4B: Design and study population: Also, important to know the percentage of attendance for the whole group of parents.

Author reply: We thank the reviewer for prompting clarification on the percentage of attendance reported. The enrolment, attendance, and program completion rates reported were reflective of all parents enrolled in the program (n=30), and the program outcomes reported were reflective of the parents that consented to participate in the research study (n=23). We have made numerous revisions to relevant sections with the hope that this is now exceptionally clear (pgs. 11, 12, 13).

Reviewer 1, Comment 5: Did the Authors treated the data as non-independent when required, because the two parents of a couple rated the same child? This point needs to be addressed.

Author reply: We thank the reviewer for bringing to our attention this very critical information that must be included. As noted in our response to comment 4A, we have now clarified that only one parent from each couple enrolled in the program was eligible to participate in the research. This decision was made to ensure there would be no dependency in the data (pg. 12).

Reviewer 1, Comment 6: Results: P 13 did the parents who didn't accept research enrollment differed from the parents who accepted (N = 23) on pre-intervention demographic characteristics? This info is important as feasibility research is being carried out.

Author reply: Unfortunately, we were only able to collect demographic data for parents that consented to the research study (n=23). We appreciate the reviewer bringing this potential study limitation to our attention, and accordingly we have discussed this comment in our limitation section (pg. 21).

Reviewer 1, Comment 7: Discussion: I found the discussion well written, clear and effective in addressing research aims, thus contributing to the literature research in a very challenging family context. I would add similar and recent research implementing eCONNECT in different context, e.g. Benzi I. et al., 2023 International Journal of Environmental Research and Public Health, 20, 3532. 1-11. <https://doi.org/10.3390/ijerph20043532>, for a more complete discussion

Author reply: We thank the reviewer for this suggestion, and we have added references for studies specifically evaluating eConnect, including the study suggested by the reviewer (pg. 18 and 20).

Reviewer 1, Comment 8: References: Some references need to be checked e.g the year of Lin, B., & Moretti, M. M. (2022; in press) eConnect: Implementation and preliminary evaluation of a virtually delivered attachment-based parenting intervention during COVID-19. Attachment & Human Development

Author reply: We appreciate the prompt to review our reference list. We have thoroughly reviewed all the references to ensure they are correct and up-to-date.

Reviewer 1, Comment 9: To sum up my comments, I found the topic addressed of great value and the paper interesting and with minor revisions for publication.

Author reply: We appreciate the suggestions made by the reviewer. We believe the manuscript has been strengthened as a result.

Reviewer 2, Comment 1: Thank you for providing me with the opportunity to review your feasibility study of eConnect for forcibly displaced parents of adolescents in Sweden. Findings from this study will offer important lessons for implementing culturally appropriate and acceptable digital parenting interventions for forcibly dislocated families. The manuscript is well written. I only have a few minor comments.

Author reply: We thank the reviewer for noting some of the strengths of the manuscript.

Reviewer 2, Comment 2: Minor proof-reading needed: for instance, on page 4, line 33, the word "period" had been written twice in the same sentence.

Author reply: We thank the reviewer for catching this typo. We have revised this typo, and the manuscript has been carefully proof-read again by all authors.

Reviewer 2, Comment 3A: Data collection: On page 9, line 6, the authors state that most parents came to the municipality to complete assessments. Where was the data collected for the parents who did not come to the municipality and who assisted this data collection?

Author reply: As suggested, we have included details of how the data was collected for the parents who did not complete the research assessments at the municipality and how we ensured that research support was available if needed (pg. 9).

Reviewer 2, Comment 3B: On page 9, line 15, the authors state that the focus group interviews were facilitated by the lead researcher. Please provide a bit more detail surrounding this decision. Does the lead researcher speak the local languages of the participants? If no, were there any translators to assist the process to ensure that participants understood the questions? If the focus groups were not conducted in the local languages of participants, it may also be worth acknowledging this as a limitation in the discussion (being reflexive of the culture, race, and language of the interviewers).

Author reply: As suggested, we have clarified that all focus groups were conducted in parents' native language. We have added contextual details surrounding the decision for the lead researcher to facilitate the focus group discussions (e.g., experience and language abilities; pg. 9). We have also acknowledged that the use of a translator for the Dari-speaking group could have resulted in loss of meaning of parents' descriptions and reported how we used the transcription process to assess the audio tape for accuracy, as noted by the reviewer (pg. 9).

Reviewer 2, Comment 4A: On page 12, line 20, the authors describe the effect size used in the analysis. It would be beneficial for interpretation of the results if 95% confidence intervals were also included.

Author reply: As suggested, we have included the 95% confidence intervals (pgs. 16-17).

Reviewer 2, Comment 4B: It would be helpful to understand how many of the 23 parents were participating with their partner, i.e., in a couple. The authors should consider how this might affect 1) the interpretation of engagement results and 2) their analysis of change. That is, it may not be enough to merely use ANOVAs to determine within-subject change when individuals are nested within a couple/family.

Author reply: We thank the reviewer for raising this very important point that must be clarified. We have clarified that one parent from each couple was eligible to participate in the study to avoid any dependency in the data (pg.12), as noted in our response to Reviewer 1 (comment 4A and 4B).

Reviewer 2, Comment 5: Outcome measures: Have the primary and secondary outcomes been tested and/or in refugee populations?

Author reply: If we understood the reviewer's question correctly, neither the Brief Child and Family Phone Interview nor the Adolescent Attachment Anxiety & Avoidance Inventory has been used or validated with refugee populations. We have made this clearer in the limitation section (pg. 21). Although, we note that the internal consistencies for these questionnaires showed acceptable levels.